# LARGE LANGUAGE MODELS FOR EXPLAINABILITY IN MACHINE LEARNING

## ABSTRACT

We investigate the potential of large language models (LLMs) in explainable artificial intelligence (XAI) by examining their ability to generate understandable explanations for machine learning (ML) models. While recent studies suggest that LLMs could effectively address the limitations of traditional explanation methods through their conversational capabilities, there has been a lack of systematic evaluation of the quality of these LLM-generated explanations. To fill this gap, this study evaluates whether LLMs can produce explanations for ML models that meet the fundamental properties of XAI using conventional ML models and explanation methods as benchmarks. The findings offer important insights into the strengths and limitations of LLMs as tools for explainable AI, provide recommendations for their appropriate use, and identify promising directions for future research.

## 1 INTRODUCTION

The rapid advancement of artificial intelligence (AI) has led to the widespread use of complex machine learning (ML) models across various sectors, raising concerns about the opacity of these "black box" systems, particularly in fields like finance, healthcare, and law (Adadi & Berrada, 2018). In response, explainable AI (XAI) has emerged to help humans understand how and why ML models make decisions. However, despite progress in XAI, there remains little consensus on how to measure explanation effectiveness, and many conventional methods still require technical expertise, making them inaccessible to non-experts (Doshi-Velez & Kim, 2017; Lockey et al., 2021).

Large language models (LLMs), such as GPT-4 and LLaMA 3, have revolutionised natural language processing and shown promise in generating explanations that are better understood by non-technical users (Brown, 2020; Liu et al., 2023). Embedding these LLMs in applications like ChatGPT and Copilot has demonstrated their potential beyond NLP, including coding and mathematical reasoning (Chang et al., 2024). Researchers have begun exploring how LLMs could be used to produce explanations for ML models, suggesting that their conversational capabilities could address the barriers associated with traditional XAI methods (Susnjak, 2023; Mavrepis et al., 2024).

While recent studies suggest that LLMs can be used to enhance XAI, they do not systematically evaluate the quality of LLM-generated explanations. This research addresses that gap by evaluating LLM-generated explanations against established XAI criteria, such as accuracy, fidelity, and stability, using conventional ML models and XAI methods as benchmarks. It provides insights into the strengths and limitations of LLMs as explainers and contributes to broader discussions about the need for rigorous evaluation within XAI.

This paper is structured as follows: The Literature Review synthesises key concepts from XAI and LLMs, reviewing their applications and highlighting the need for a more rigorous evaluation of LLMs as explainers of ML models. The Methodology outlines the experimental design used to assess LLM-generated explanations, detailing the selection of XAI properties, ML models, benchmarks, and evaluation metrics. The Results present the performance of LLMs in generating explanations, comparing them to traditional methods. Finally, the Discussion and Conclusion address the research's limitations, broader implications, and potential future directions for LLMs in XAI. Additional details on the experimental methodology, results, readability measures, and prompts employed are provided in the appendices A.1, A.2, A.3 and A.4.

## 2 LITERATURE SURVEY

While significant research underpins the fields of XAI and LLMs, their intersection has received much less attention. To lay the theoretical foundation for exploring this, this review tackles their relevant background and examines recent research that applies LLMs in XAI contexts.

### 2.1 EXPLAINABILITY IN AI

There is growing concern about the lack of understanding of AI system decision-making (Adadi & Berrada, 2018), with research identifying several sources of these concerns: adversarial attacks (Akhtar & Mian, 2018; Chen et al., 2017), algorithmic aversion (Dietvorst et al., 2015), complexity (Došilović et al., 2018; Du et al., 2019), discriminatory bias (Mehrabi et al., 2021) and legal requirements (Goodman & Flaxman, 2017; MacCarthy, 2019; Gursoy et al., 2022). These concerns have driven extensive research into improving how humans understand AI systems. However, the complexity and ambiguity of explainability have sparked considerable debate over its definitions and approaches, highlighting the need for a more formalised framework.

Evaluating the potential of LLMs in XAI requires a clear understanding of what "explainability" entails. However, research in XAI has been hindered by a lack of consensus on fundamental definitions (Doshi-Velez & Kim, 2017; Murdoch et al., 2019; Rosenfeld & Richardson, 2019) and the conflation of terms such as transparency, interpretability, and explainability (Arrieta et al., 2020; Došilović et al., 2018; Linardatos et al., 2020). *Transparency* refers to the inherent ability of an AI system's inner workings to be understood by humans (Belle & Papantonis, 2021; Lipton, 2018; Arrieta et al., 2020) and *Interpretability* refers to the ability of human's how an AI system produced its decision (Biran & Cotton, 2017; Molnar, 2024; Belle & Papantonis, 2021; Doshi-Velez & Kim, 2017; Gilpin et al., 2018; Minh et al., 2022). Explainability differs as it refers to an AI system's ability to make its functionality understandable to a specific audience by providing social knowledge exchanges (explanations), framed by explainer and recipient beliefs (Arrieta et al., 2020; Miller, 2019).

Determining the suitability of LLMs for XAI requires a critical evaluation of explanation quality. However, as (Confalonieri et al., 2021) stresses, there is little consensus within XAI research on what constitutes a good explanation so it is crucial to explore the various properties of explainability. Phillips et al. (2021) outlines four system-level explainability principles for an AI system:

1. The system produces or is accompanied by explanations.
2. The explanations are meaningful to their intended human audience.
3. The explanations accurately represent the system's inner workings.
4. The explanations communicate when the system operates outside its intended limits.

Several authors such as (Belle & Papantonis, 2021; Craven & Shavlik, 1999; Molnar, 2024; Robnik-Šikonja & Bohanec, 2018; Nauta et al., 2023) have proposed explanation-level properties. However, Miller (2019), building on Confalonieri et al. (2021), critiques such approaches as overly technical and draws upon social sciences research to advocate for more human-centered properties. Combining these offers a holistic perspective on effective explanations:

*Comprehensibility*: How comprehensible the explanation is to a human.

*Fidelity*: How accurately the explanation captures the model's behaviour.

*Accuracy*: The ability of explanations to predict novel samples.

*Scalability*: How well the explanatory method scales with input data and model complexity.

*Generality*: How applicable the explanatory method is to different models.

*Consistency*: The similarity between explanations of different models trained on the same task with similar predictions.

*Stability*: The similarity of explanations for instances in the same local input area.

*Certainty*: Whether the explanations reflect the model's output confidence.

*Novelty*: Whether the explanation can identify novel instances unseen during training.

*Degree of importance*: How well the explanation reflects feature influence on model decisions.

*Representativeness*: The extent of the model's instances that the explanation covers.

*Completeness*: How well the explanatory method explains the entirety of the model's decision.

*Social*: Explanations are social interactions and framed by the explainer's and recipient's beliefs.

*Contrastive*: Effective explanations are often framed as counterfactuals.

*Selective*: Effective explanations focus on the most influential features, not detailing each factor.

*Causal*: Effective explanations focus on causal reasons, not probabilities.

*Semantic*: Symbolically represented explanations can be better tailored to their target audience.

While these properties clarify what constitutes an effective explanation there is also a lack of consensus on the evaluation of explainability methods (Murdoch et al., 2019; Du et al., 2019). Furthermore, Doshi-Velez & Kim (2017) argues that what evaluation does occur lacks systematic rigour, relying on the model's inherent transparency or assuming explainability if the model meets performance expectations. To address this, they propose an explanation evaluation task taxonomy: *application-grounded evaluation* (real-world human task performance), *human-grounded evaluation* (simplified human task performance), and *functionally-grounded evaluation*, (evaluation against definitions of explainability). Phillips et al. (2021) suggest a different perspective on evaluating explanatory methods, involving *evaluating explanation accuracy* and *evaluating explanation meaningfulness*.

The approaches advocated by Doshi-Velez & Kim (2017) and Phillips et al. (2021) can be summarised as either *performance-focused*, where explanations are indirectly assessed by seeing if they improve human performance in a real-world or simplified task, or *explanation-focused*, where explanations are directly compared against another explanation, either using human evaluation or evaluated against a formal definition or criteria. Accuracy and meaning are both crucial because meaningful explanations build trust in AI systems (Logg et al., 2019; Shin, 2021) while a lack of accuracy and robustness significantly harms human trust them (Dietvorst et al., 2015; Ahn et al., 2024).

Angelov et al. (2021) outline three explainability methods characteristics: Usage, which refers to whether the method is model-specific or model-agnostic; scope, which defines whether the method applies globally to the entire model or locally to a subset; and methodology, which specifies the part of the system addressed, such as inputs or features. Additionally, Belle & Papantonis (2021), referencing Arrieta et al. (2020), categorise explanatory methods by their outputs into four main types: explanations by example, local explanations, simplification methods, and feature relevance methods. These methods can produce various outputs, including textual (natural language), visual (charts and graphs), and numerical explanations that quantify relationships between model components.

An exhaustive list of explainability methods is beyond the scope of this paper. Instead, a brief overview of three widely used model-agnostic, local methods employed in this study will be provided: *Local Interpretable Model-Agnostic Explanations (LIME)* simplifies complex models by locally approximating them with transparent models such as linear models or decision trees, using their properties to explain the complex model (Ribeiro et al., 2016). LIME benefits from strong theory and quantifiable fidelity but suffers from instability and difficulty in defining local areas (Molnar, 2024). *SHapely Additive exPlanations (SHAP)* uses game theoretically optimal Shapely values to calculate the average expected marginal contribution of each feature (Lundberg & Lee, 2017). SHAP benefits from its roots in game theory and high fidelity and completeness. However, it is computationally complex, ignores feature independence and can be misled by perturbations (Molnar, 2024). *DiCE Counterfactuals* produce contrastive explanations by finding the minimal changes necessary to change an input example's predicted output (Wachter et al., 2017). Counterfactuals provide understandable explanations that do not require access to the underlying data or model, making it suitable where data protection concerns are essential. However, many counterfactuals can be generated for the same input and there is no straightforward method to identify which is best (Molnar, 2024).

## 2.2 LARGE LANGUAGE MODELS

LLMs are computational systems that have become an important tool for natural language processing (NLP). Fundamentally, an LLM aims to predict the following sequence of words given a specified input sequence (Min et al., 2023).

Understanding how LLM performance is evaluated provides crucial context to their capabilities. Chang et al. (2024) outline an evaluation taxonomy that addresses three key questions: what, where, and how to evaluate LLMs. The "what" refers to task selection, involving traditional NLP tasks and newer domains like mathematics, law, and healthcare. The "where" focuses on standard benchmarks, which consist of a problem statement, a representative dataset, and performance metrics. Finally, the "how" involves automated computational approaches or human assessment.

LLMs were first evaluated on standard NLP tasks and benchmarks and demonstrated state-of-the-art capabilities (Brown, 2020; Liu et al., 2023; Chang et al., 2024). LLMs have also demonstrated performance in areas outside of NLP. LLMs have exhibited the ability to reason mathematically with GPT-4 able to tackle undergraduate-level problems (Bubeck et al., 2023; Frieder et al., 2024). In engineering, LLMs can generate computer code and have been integrated into products such as Github Co-Pilot (Bubeck et al., 2023; Dakhel et al., 2023; Nguyen & Nadi, 2022). In education, LLMs can support learners and teachers in educational tasks (Jeon & Lee, 2023). LLMs have demonstrated performance in medical tasks such as passing licensing exams, clinical reasoning, and record analysis (Shen et al., 2023; Singhal et al., 2023; Thirunavukarasu et al., 2023; Yang et al., 2022). Choi et al. (2021) also demonstrated that ChatGPT can pass university-level law exams, and Lu & Wong (2023) showed that ChatGPT can perform tasks done by litigation lawyers.

Despite their impressive capabilities, LLMs have been demonstrated to have several shortcomings, calling into question their performance: misleading or non-sensical information known as *hallucations* (Ji et al., 2023); *Adversarial examples*, where minor alterations to prompts significantly alter outputs (Zhu et al., 2024); *Misuse* such as fraud, misinformation, or plagiarism Brown (2020); Khalil & Er (2023); Meyer et al. (2023); Shen et al. (2023); *Bias* such as occupational, gender, and ethnic (Brown, 2020; Ray, 2023); *Toxicity*, where LLMs can be coerced into producing responses and adopting personas that exhibit harmful stereotypes (Deshpande et al., 2023; Liu et al., 2023); *Energy usage* since training LLMs require high energy consumption, raising concerns about their environmental impact (Brown, 2020; Ray, 2023; Touvron et al., 2023).

Research has focused on three strategies to address these issues: prompt engineering, contextual examples, and fine-tuning. *Prompt engineering* involves crafting natural language inputs to achieve desired outputs (Denny et al., 2023; White et al., 2023; Zhou et al., 2023). *Contextual examples* enhance performance by shifting LLMs from zero-shot to few-shot settings, although their effectiveness varies based on the number and sequence of provided examples (Brown, 2020; Liu et al., 2021). *Fine-tuning* applies supervised learning to small, representative datasets to improve task-specific performance, allowing users to enhance LLM capabilities for tasks such as code generation or medical literature analysis (Radford et al., 2018; Chen et al., 2021; Wu et al., 2023).

## 2.3 APPLYING LLMS TO XAI

While there have been studies that explore the use of conversational agents or interfaces for XAI, such as those developed by Kuźba & Biecek (2020), Nguyen et al. (2022), and Guimaraes et al. (2022), only nine studies could be identified as exploring LLMs for this task at the time of writing. Susnjak (2023) used ChatGPT to generate natural language explanations from the outputs of model predictions, SHAP, and counterfactuals for individual instances in a learning analytics context, however, no evaluation of explanation effectiveness was performed. Ali & Kostakos (2023) developed a cybersecurity anomaly detection system using random forests, with SHAP and LIME outputs fed into ChatGPT with selected instances to generate natural language explanations. Yang et al. (2023) applied ChatGPT to extract, analyse, and explain digital advertising samples, which were evaluated by surveying 12 professionals who provided high-level positive feedback. Guo et al. (2024) developed a fine-tuned LLM to forecast traffic flow and generate explanations, which ChatGPT subsequently summarised, while the LLM performed well at forecasting, the explainability of the evaluations was not evaluated. Nazary et al. (2024) compared clinical predictions generated by ChatGPT against conventional ML models by evaluating the prediction accuracy, with the ML models being more accurate in most settings. Serafim et al. (2024) used ChatGPT to produce explanations of the outputs of a decision tree trained on the Iris dataset, providing guidance on prompt construction and a brief subjective evaluation of the explanations. Silva et al. (2024) received positive user feedback when using ChatGPT as a movie recommender system where generated recommendations were compared against random recommendations from popular movie lists by surveying participants. Maddigan et al. (2024) used ChatGPT to generate natural language explanations of genetic

programming trees. Mavrepis et al. (2024) trained a custom LLM using ChatGPT to generate explanations of SHAP, LIME, and GradCAM outputs. Prompt engineering and contextual information were used to enhance explanations. Surveyed professionals found the explanations understandable.

Many authors highlight the key benefit of LLMs for XAI as using their conversational capabilities to produce more understandable and accessible natural language explanations. This is appealing because many XAI methods require substantial expertise to understand, making communicating their results to non-technical users difficult (Maddigan et al., 2024). However, no study evaluated explanation effectiveness against established XAI properties using evidence-based methodologies, like those described by Doshi-Velez & Kim (2017); Phillips et al. (2021); Ji et al. (2023).

## 3 METHODOLOGY

Research in XAI faces significant challenges due to the lack of agreed-upon methodologies and metrics. This issue is particularly evident in research into the application of LLMs for XAI, where studies (Serafim et al., 2024; Guo et al., 2024; Maddigan et al., 2024) demonstrate LLMs' explanatory capabilities but lack a rigorous assessment of performance and robustness. We addresses that gap through an experimental framework that evaluates LLMs against established properties of XAI.

### 3.1 RESEARCH DESIGN

We aim to answer the question: **Can LLM-generated explanations satisfy established properties of XAI?**

We adopt a quantitative *explanation-focused* approach, using the functionally grounded approach outlined by Doshi-Velez & Kim (2017); Phillips et al. (2021), to evaluate LLM explanation quality. LLM-generated explanations were compared to explanations from conventional XAI methods. This allows for a more objective assessment of LLMs' explanatory capabilities against a clear set of quantifiable properties. An experimental framework was designed to evaluate the selected XAI properties systematically. Framework development included:

*Property selection*: Choosing the XAI properties for evaluation.

*LLM selection*: Selecting the accessible LLMs representative of their modern capabilities.

*Task selection*: Choosing commonly used datasets, representative of real-world tasks.

*Machine learning model selection*: Selecting applicable models often used in research and industry.

*LLM and benchmark explanations*: Choosing explanation types representative of real-world task requirements and produced by conventional explanatory methods.

**Property Selection**   Since explanations are inherently complex and involve various dimensions of quality, selecting a broad range of properties was essential for a holistic evaluation. However, given the lack of consensus in properties, our approach involved synthesising the properties specified by Belle & Papantonis (2021); Craven & Shavlik (1999); Molnar (2024); Robnik-Šikonja & Bohanec (2018); Nauta et al. (2023) as detailed in the section 2.1 of the literature review. We selected ten properties from those detailed based on their specificity, quantifiability and suitability for a functionally grounded approach. Additionally, to address the tendency of LLMs to generate nonsensical outputs, the robustness property was defined to measure the frequency of errors in LLM-generated explanations. The full list of properties we selected are *Accuracy*, *Selectivity*, *Fidelity*, *Completeness*, *Contrastness*, *Certainty*, *Degree of Importance*, *Consistency*, *Stability*, *Robustness*, and *Comprehensibility*.

**LLM Selection**   We selected a sample of six LLMs based on their prominence, capabilities, and accessibility to ensure a representative sample of the latest modern LLMs available. While not all of the latest models could be included due to cost and availability limitations, these six capture a range of LLM developers, architectures and sizes:

1. GPT Models by OpenAI: gpt-4o-mini, a smaller, resource-optimised version of the latest GPT-4o model, and gpt-4-turbo, the largest model from the previous generation.

2. LLaMA 3 models by Meta: llama3-70b-8192, the largest and intended for large-scale applications, and llama3-8b-8192, the smallest and intended for small-scale applications.

3. Gemma models by Google: Gemma2-9b, a medium-sized model from the latest Gemma generation, and Gemma-7b, the largest model from the first generation.

**Task Selection**  Selecting appropriate tasks is crucial for evaluation design (Chang et al., 2024). To ensure a broadly representative and thorough comparison, we selected two standard ML predictive problems:

- Classification on the Adult dataset (Becker & Kohavi, 1996).
- Regression on the California Housing dataset (Pace & Barry, 1997).

as we expect their popularity and simplicity to allow for a better exposition of LLMs' explainability properties. Due to the resource constraints of querying LLMs, a 99% to 1% train-test split ratio was used with a fixed random seed, resulting in 261 and 207 test samples for the Adult Income and the California Housing datasets respectively.

**Model Selection and Training**  Five machine learning models were selected for each task to represent commonly used models with varying architectural complexity and levels of interpretability. For the Adult Income classification task, Logistic Regression, Decision Tree, Random Forest, KNN, and Gradient Boosted Tree were chosen, while for the California Housing regression task, Linear Regression, Decision Tree, Random Forest, KNN, and Gradient Boosted Tree were selected. Each model's hyperparameters were tuned using cross-validation on the training sets, and the best hyperparameters were used to train the models, which were then saved to generate predictions for the experiments. This selection of models allows for the evaluation of LLM explanations across a diverse range of architectures and interpretability levels.

**Explanatory Method Selection**  The explanatory methods were chosen based on four criteria: range of outputs, complementary pairing with conventional methods, applicability across various machine learning models, and established use in XAI research. Each LLM-generated explanation type was paired with a similar benchmark method (e.g., DiCE counterfactuals) to enable a like-for-like comparison. The explanation types selected for LLM generation included predictions, predicted probabilities, most influential features, feature importance values, linear coefficients, marginal feature contributions, counterfactuals, and natural language explanations. Conventional benchmarks such as ML model predictions and predicted probabilities, DiCE counterfactuals, LIME, linear model coefficients, and SHAP values were used as comparisons. Although predictions are not typically viewed as explanatory methods, assessing LLMs' predictive capabilities is crucial because predictions reveal a model's inner workings (Lipton, 2018; Phillips et al., 2021).

## 3.2  LLM EXPLANATION COLLECTION

This section details how the LLM-generated explanations were collected from the six LLMs used across various tasks and explanation types. This process involved selecting tools and technologies, designing effective prompts, and API querying.

To efficiently collect LLM-generated explanations, APIs were used, with the OpenAI API querying GPT models and the Groq Cloud API accessing LLaMA and Gemma models, both sharing a common framework for consistent querying. Python, along with its data science libraries like Pandas, was employed for efficient data collection and manipulation. The LLM-generated explanations for each task were stored in CSV files, LLM, and ML model combinations to facilitate easy analysis.

A standardised query structure was implemented across the OpenAI and Groq Cloud APIs, ensuring uniformity in the API query format. Batch processing was used to address systematic errors, such as incorrect formatting or an incorrect number of responses, which were more frequent with larger data samples; consequently, input data was divided into smaller batches of 25 samples or fewer to minimise these errors. Error handling and data cleaning were also performed to detect and correct formatting mistakes, with error frequency reviewed as part of the robustness analysis. Finally, prompts were modularised, following a standard structure that could be tailored to the specific explanation type, task, and machine learning model.

Each explanation type required the construction of a unique prompt. The querying structure for both ChatGPT and Groq Cloud APIs was identical, allowing each prompt to be reused across LLMs. The APIs support two types of messages: the system role, which provides contextual information to guide the model's behaviour, and the user role, which represents human input to elicit responses from the LLM. This structure enables the initialisation of LLMs with contextual information before issuing specific instructions. A modular prompt approach was employed, following best practices such as specifying output formats and constraints. The prompts were divided into four components: role context, defining the LLM's role and objectives; data context, describing the dataset's features and target labels; input context, providing sample data; and prompt context, instructing the LLM on the desired output format and constraints. These components were then submitted in a single API query, batched together through a Python list of dictionaries. See A.3 for details.

## 4 RESULTS

This section presents the findings of the experiments devised to evaluate the capabilities of LLMs for XAI. Each experiment assesses how the six LLMs performed against one of eleven selected properties of XAI using quantitative measures and conventional methods as benchmarks. The full results of each experiment are detailed in A.2. Summaries of the results are displayed in table 1 and table 2, with the values of the best-performing LLMs highlighted in each row:

**Accuracy**  was measured by evaluating the predictions made by each LLM to the actual test set labels using the accuracy measure for the Adult Income dataset and the root mean square error (RMSE) for the California Housing dataset. This process was repeated for each dataset's ML model, which served as benchmarks. In the Adult Income task, all LLMs, except gpt-4-turbo, underperformed each conventional machine learning model. Larger LLM models generally performed better, achieving an accuracy score similar to the scores of the decision tree and logistic regression models. In the California Housing task, all LLMs underperformed compared to conventional models, with larger LLMs faring better. However, error rates were much higher, with the best LLM, gpt-4-turbo, having an RMSE nearly three times that of the worst conventional model.

**Selectivity**  was measured by comparing the LLM-generated explanations for each ML model with those from the DiCE Counterfactual method for the respective task using cosine similarity. Higher mean cosine similarity indicated greater selectivity in identifying influential features. In both the Adult Income classification and California Housing regression tasks, LLM-generated explanations showed varying similarity to DiCE counterfactuals. The LLaMA models had the highest selectivity for the Adult Income task, though only around 0.29. In the California Housing task, LLMs showed higher selectivity, with gpt-4-turbo and gemma2-9b performing best, while smaller models, like gpt-4o-mini, underperformed, showing significant misalignment.

**Fidelity**  was evaluated by comparing the LLM-generated estimations of the coefficients of logistic regression and linear regression models with their actual coefficients using cosine similarity. On both tasks, the LLMs struggled to identify the correct coefficients of either linear model, with most LLMs exhibiting low fidelity scores. The best-performing models were llama3-70b-8192 (0.28) for the Adult Income task and gpt-4-turbo (0.57) for the California Housing task. However, the results show high variability, with gpt-4o-mini and gemma* displaying negative similarity scores, indicating poor alignment with the actual model coefficients.

**Completeness**  was evaluated by comparing the LLM-generated estimations of each feature's marginal contributions to model predictions with their corresponding SHAP values using cosine similarity. The completeness of each LLM was quantified by calculating the mean cosine similarity for each of the LLM's estimations, with a higher cosine similarity indicating higher completeness. On both tasks, the results show that each LLM exhibited low negative average cosine similarity scores for each task, suggesting that LLMs explanations lack completeness.

**Constrastness**  was measured by assessing LLM-generated counterfactuals' ability to change model decisions in the specified manner. The Adult Income counterfactuals were evaluated by calculating accuracy based on the model's new prediction compared to the intended change, e.g., if

the original label was 0, the counterfactual's target was 1. The California Housing counterfactuals aimed to increase the original median house price by between 20% and 40% and were measured by calculating the average percentage change, with a target range of 0.20 to 0.40. The results show that LLM-generated counterfactuals struggled to consistently change model decisions. In the Adult Income task, gpt-4-turbo had the highest accuracy (0.27), while other models ranged from 0.23 to 0.24, highlighting difficulties in generating effective counterfactuals. For the California Housing task, gemma-7b was the only LLM to meet the target range of 0.2 to 0.4, with llama3-8b-8192 and gpt-4-turbo performing the worst.

| Property | gpt-4-turbo | gpt-4o-mini | llama3-70b | llama3-8b | gemma-7b | gemma2-9b |
|---|---|---|---|---|---|---|
| Accuracy | **0.78** | 0.67 | 0.55 | 0.69 | 0.55 | 0.74 |
| Selectivity | 0.24 | -0.03 | **0.29** | **0.29** | 0.24 | 0.13 |
| Fidelity | -0.10 | -0.07 | **0.28** | 0.07 | -0.21 | 0.08 |
| Completeness | -0.13 | -0.19 | -0.15 | -0.18 | **-0.03** | -0.19 |
| Contrastness | **0.27** | 0.24 | 0.23 | 0.23 | 0.24 | 0.23 |
| Certainty | 0.36 | 0.48 | **0.60** | 0.48 | 0.58 | 0.39 |
| Deg. of Importance | 0.03 | -0.01 | -0.02 | -0.01 | **0.11** | 0.02 |
| Consistency | 0.81 | 0.77 | **0.84** | 0.81 | 0.80 | 0.69 |
| Stability | 0.64 | **0.99*** | 0.62 | 0.47 | 0.57 | 0.57 |
| Robustness | **0.0** | 35.7 | **0.0** | **0.0** | 3.3 | 0.1 |
| Comprehensibility | 11.7 | 10.1 | 8.8 | 9.5 | **7.6** | 12.1 |

Table 1: **Adult Income Classification: Accuracy**, **Contrastness**, and **Robustness** are normalised from 0 to 1. **Selectivity**, **Fidelity**, **Completeness**, **Deg. of Importance**, **Consistency**, and **Stability** are based on cosine similarity metric. **Certainty** is based on an RMSE measurement. **Comprehensibility** is based on the Flesch-Kincaid readability score (lower means simpler).

| Property | gpt-4-turbo | gpt-4o-mini | llama3-70b | llama3-8b | gemma-7b | gemma2-9b |
|---|---|---|---|---|---|---|
| Accuracy | **199,891** | 320,902 | 261,863 | 317,304 | 273,450 | 295,339 |
| Selectivity | **0.43** | 0.39 | 0.38 | 0.38 | 0.34 | **0.43** |
| Fidelity | **0.57** | -0.36 | 0.23 | 0.03 | -0.63 | -0.60 |
| Completeness | -0.04 | -0.03 | -0.03 | -0.05 | -0.08 | **-0.02** |
| Contrastness | 0.73 | 0.64 | 0.51 | **0.76** | 0.37 | 0.48 |
| Certainty | 0.36 | 0.48 | **0.60** | 0.48 | 0.58 | 0.39 |
| Deg. of Importance | -0.13 | -0.08 | -0.07 | -0.10 | -0.20 | **-0.05** |
| Consistency | 0.95 | 0.94 | 0.91 | **0.97** | 0.85 | 0.93 |
| Stability | 0.81 | **0.92** | 0.89 | 0.89 | 0.87 | 0.87 |
| Robustness | **0.0** | 0.5 | **0.0** | 6.0 | **0.0** | 0.8 |
| Comprehensibility | 12.7 | 10.7 | 10.5 | **8.2** | **8.2** | 12.4 |

Table 2: **California Housing Regression: Accuracy**, and **Certainty** is based on an RMSE measurement. **Contrastness**, and **Robustness** are normalised from 0 to 1. **Selectivity**, **Fidelity**, **Completeness**, **Deg. of Importance**, **Consistency**, and **Stability** are based on cosine similarity metric. **Comprehensibility** is based on the Flesch-Kincaid readability score (lower means simpler).

**Certainty** was evaluated by comparing LLM estimates of the probabilities of predicted class labels to the class probabilities produced by the classifier models on the Adult Income dataset. The experiment used the RMSE to calculate the error between the LLM estimates and the actual model probabilities. The LLM's certainty metric was the mean RMSE across all samples and models. The results show that all LLMs exhibited high error rates in estimating class probabilities. Larger models performed better, however, high RMSE scores across all LLMs indicate difficulty in accurately estimating class probabilities. Even the better-performing models had RMSE scores between 0.36 and 0.39, while the worst models exceeded 0.5, indicating low confidence in their predictions.

**Degree of importance** was evaluated by comparing the feature importance values generated by LLMs with those derived from LIME using cosine similarity. The degree of importance score for

each LLM was determined by calculating the mean cosine similarity across all models, where a higher cosine similarity indicates a better alignment with the degree of importance property. The results for both tasks show significant misalignment between the LLM and LIME estimations of feature importance. Larger LLMs performed better on the classification task, while performances were more varied on the regression task. Additionally, the negative scores on the regression task indicate significant misalignment with LIME.

**Consistency** was evaluated by calculating the cosine similarity of the most influential features identified by the LLMs for the same samples across each ML model for both tasks. The mean cosine similarity was computed for each LLM across each model as the consistency metric. The results show that LLMs generated consistent explanations between models across both tasks, as measured by mean cosine similarity. he larger LLMs also generally exhibited higher consistency over smaller models.

**Stability** of the explanations was evaluated for each of the LLMs. To assess this, identical samples were generated, shuffled, and reindexed. The LLMs were then tasked with identifying the most influential features for each sample. Stability was quantified by calculating the cosine similarity of the most influential features across the identical samples. This process was repeated for each model, and the mean cosine similarity scores for each LLM were computed. Most LLMs showed significant variation in identifying the most influential features. Smaller models generally performed worse, but gpt-4o-mini had unusually high stability scores because it incorrectly marked every feature as most influential, ignoring the instructions.

**Robustness** refers to the ability of the LLM-generated explanations to be error-free and was evaluated using two criteria: the percentage of explanations generated by LLMs that had an incorrect number of rows or columns and the rate of explanations that failed to adhere to instructions by either returning all features when a specific selection was required or outputting all zeroes or NaN values when features needed to be quantified. The results showed that LLMs frequently returned the wrong number of rows or columns, even when prompts were clear. Row errors were more common, with LLMs more likely to return an incorrect number of samples. The LLMs also often failed to follow instructions, resulting in invalid outputs. The larger models had the lowest rates of invalid outputs, with zero errors on both tasks.

**Comprehensibility** was measured by evaluating the LLM-generated natural language explanations using standard readability metrics such as Flesch and Flesch-Kincaid, which measure factors like sentence length and word difficulty as a proxy for comprehensibility (Ley & Florio, 1996; Kincaid & Delionbach, 1973; Eltorai et al., 2015; Spache, 1953). The results show that the LLM-generated explanations exhibit a moderate level of readability, accessible to readers with high school graduate or college reading levels. Larger models typically produced explanations requiring more advanced reading levels, while smaller models produced more straightforward explanations.

## 5 DISCUSSION

Our work examines the ability of LLMs generate explanations that align with eleven fundamental properties of XAI (Belle & Papantonis, 2021; Molnar, 2024; Nauta et al., 2023). Using a functionally grounded approach (Doshi-Velez & Kim, 2017), the study quantitatively assesses six prominent LLMs across various ML models and tasks, comparing their explanations to conventional XAI methods. The findings provide a profile of LLM explanations with specific strengths and weaknesses.

We attempted to provide a broad assessment of LLMs in XAI. However, we recognise three primary limitations to this work: resource constraints, dependencies on prompt construction, and the lack of standardised benchmarks and metrics. LLMs are computationally and financially intensive (Brown, 2020; Ray, 2023), making it necessary to limit the scope of this study to affordable LLMs applied on 1% of the test sets. While we followed generally accepted prompt engineering guidelines (Wei et al., 2022; Chen et al., 2023), this work does not focus on prompt engineering for improved explanations, unlike Zhao et al. (2021). Since there are no standard ground-truth benchmarks for XAI, we compare against an ensemble of established methods (Phillips et al., 2021) that target similar requirements.

Effective explanations in XAI should be selective and contrastive, highlighting features that significantly impact model decisions. However, compared to methods like the DiCE counterfactuals (Wachter et al., 2017), LLMs struggled to identify the influential features of model outputs. Similarly, the LLMs could not consistently generate effective counterfactuals. These limitations suggest that LLMs currently lack the capacity to understand the influence of input features, rendering them unsuitable for reliable feature-based explanations in XAI. Furthermore, understanding and accurately reflecting a model's decision-making process is critical in XAI (Miller, 2019; Arrieta et al., 2020; Belle & Papantonis, 2021). However, fidelity and completeness experiments showed that LLMs struggled to capture the underlying mechanisms of model decisions. Furthermore, the LLMs struggled to recognise linear model coefficients and identify feature marginal contributions, especially compared to SHAP values (Lundberg & Lee, 2017). This lack of comprehension further limits their applicability to XAI, demonstrating they cannot convey the actual workings of ML models. Another significant aspect of XAI is a method's ability to highlight the importance of a model's confidence in its decisions (Phillips et al., 2021; Molnar, 2024). While LLMs produced reasonably accurate classification predictions, they underperformed in comparison to simpler models, particularly in regression tasks and classification probability estimates. Trust in AI systems depends on the consistency and reliability of the explanations provided (Lockey et al., 2021). The stability and robustness experiments demonstrated that LLM-generated explanations are highly volatile, even when presented with identical inputs. They also revealed frequent basic errors and an inability to follow explicit instructions, an observation consistent to earlier work (Zhao et al., 2021; Ji et al., 2023). This lack of stability and robustness reduces the practical usability of LLMs in XAI and risks eroding trust, as erroneous outputs undermine confidence in AI systems (Dietvorst et al., 2015).

Despite these limitations, LLMs show some promise in acting as post-hoc explainers (Belle & Papantonis, 2021; Molnar, 2024; Arrieta et al., 2020). The comprehensibility experiment showed that LLMs can generate explanations accessible to educated audiences. Unlike traditional explanatory methods that often use technical jargon, LLMs present model outputs in a more understandable way for non-technical stakeholders. This can expand the impact of ML models by making their decision-making processes clearer to a broader audience.

Several critical directions remain for future research in improving LLM-generated explanations. A major challenge is the lack of standardised benchmarks for evaluation (Bodria et al., 2023; Sithakoul et al., 2024). Additionally, enhancing LLM performance can be approached through three key avenues: (i) refining prompt engineering methods could lead to more accurate, consistent, and contextually relevant explanations (Maddigan et al., 2024). (ii) leveraging contextual examples, where providing input-explanation pairs may guide LLMs in generating more meaningful insights, as suggested by Nazary et al. (2024). (iii) finetuning LLMs on domain-specific datasets, as demonstrated in studies such as Radford et al. (2018) and Wu et al. (2023), holds significant potential for improving task-specific explanation generation. Another direction would be to consider multi-modal approaches. Visual explanations may provide a more intuitive understanding of AI models (Belle & Papantonis, 2021; Arrieta et al., 2020). Moreover, LLMs can also generate and execute computer code (Bubeck et al., 2023; Nguyen & Nadi, 2022), but this study restricted these capabilities, leaving an opportunity for future research on multi-modality and XAI.

Our work demonstrates that system level requirements of XAI (Phillips et al., 2021) are not satisfied by LLMs. The explanations generated by LLMs often lack accuracy, are prone to errors, and fail to offer meaningful insights into the inner workings of models. As such, LLMs may be better suited for roles as translators rather than explainers (Susnjak, 2023). In this capacity LLMs could translate the outputs of conventional explanatory methods into more understandable formats, which could help foster trust in AI systems (Logg et al., 2019; Shin, 2021).

In conclusion, this work outlines the ability of LLMs to explain ML models by comparing them with established XAI properties. The results suggest that, at present, LLMs struggle to consistently identify important features, understand the decision-making processes of models, and produce stable, error-free explanations. As a result, conventional methods currently offer more reliable and reproducible explanations. However, LLMs show promise as post-hoc explainers, particularly due to their accessibility and potential to clearly expose explanations. This research represents the first rigorous evaluation of LLMs against XAI standards (Guo et al., 2024; Serafim et al., 2024; Mavrepis et al., 2024), and future work should aim to further evaluate LLMs across a wider range of datasets and tasks. Additionally, research should explore their potential role of LLMs as translators for conventional explanatory methods, helping bridge knowledge gaps and enhance stakeholder trust.

## 6 ETHICS STATEMENT

We did not collect data to support this research, however, there are indirect implications of supporting LLMs for XAI. The application of LLMs to XAI brings broader considerations that need to be addressed through regulatory frameworks. Privacy (Pan et al., 2020; Yao et al., 2024) and safety (Zhu et al., 2024; Brown, 2020; Meyer et al., 2023) remain open issues for LLMs. There is also a need to assess whether LLM-generated explanations are sufficient to meet legal obligations, particularly in contexts such as the EU GDPR's "right to an explanation" (Goodman & Flaxman, 2017). Lastly, an important policy consideration involves ensuring that LLM explanations are unbiased (Mehrabi et al., 2021).

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

## A APPENDIX

### A.1 EXPERIMENT DETAILS

The objective and components of each experiment are detailed below:

#### A.1.1 1. ACCURACY

*Objective*: Evaluate How well the LLMs can predict using novel samples.

*LLM Explanation Type*: Predictions.

*Benchmark*: The predictions made by each machine learning model.

*Quantitative Metric*: Accuracy score for the Adult Income classification dataset and the root mean square error (RMSE) for the California Housing regression dataset.

*Task and Model Scope*: Each dataset and machine learning model.

*Procedure*: The LLM-generated predictions were compared to the test set actual labels using the relevant metric for both datasets. This is repeated for each benchmark machine learning model. Each model was ranked by their metrics to determine overall performance.

#### A.1.2 2. SELECTIVITY

*Objective*: Assess how well LLM explanations capture the few most influential features on the model's decision.

*LLM explanation type*: Most influential features.

*Benchmark*: The features identified to be changed by the DiCE counterfactual method.

*Quantitative metric*: The mean cosine similarity between the features identified in the LLM-generated explanation and the DiCE counterfactual explanation across each task's models.

*Task and model scope*: Each dataset and machine learning model.

*Procedure*: The sample features identified for change by DiCE counterfactuals were transformed alongside LLM-generated features into a numerical format using an encoder for categorical features and standard scaling for numerical ones. Missing values are set as a standard constant of -1 to differentiate them from other values in the dataset. The cosine similarity was calculated for each feature pair across all corresponding samples in the LLM-generated explanations.

### A.1.3   3. FIDELITY

*Objective*: Assess how well LLM explanations capture machine learning model behaviour.

*LLM explanation type*: Linear coefficients.

*Benchmark*: The linear coefficients of the logistic and linear regression models.

*Quantitative metric*: The cosine similarity between the LLM-generated linear and the actual linear model coefficients.

*Task and model scope*: The linear models used in each dataset.

*Procedure*: The LLM-generated coefficients were transformed into feature vectors, and the cosine similarity between them and their respective benchmark was calculated.

### A.1.4   4. COMPLETENESS

*Objective*: Assess how well the explanatory method explains the entirety of the model's behaviour.

*LLM explanation type*: Marginal contributions.

*Benchmark*: The SHAP values for each test set sample.

*Quantitative metric*: The mean cosine similarity across the LLM marginal contributions and the SHAP values.

*Task and model scope*: Each task and model.

*Procedure*: The LLM-generated feature vectors for each sample were compared with their corresponding standard-scaled SHAP values, calculating their cosine similarity. Missing values are set as a standard constant of -1 to differentiate them from other values. The mean cosine similarity across each ML model was calculated as each LLM's completeness score.

### A.1.5   5. CONTRASTNESS

*Objective*: Evaluate the effectiveness of LLMs at generating counterfactual explanations.

*LLM explanation type*: Counterfactuals.

*Benchmark*: For the Adult Income task, DiCE counterfactuals were generated to change the original label from 0 (low income) to 1 (high income) or vice versa. For the California Housing task, DiCE counterfactuals aimed to increase the original median house price by between 20% and 40%.

*Quantitative metric*: The mean counterfactual accuracy was measured for the Adult Income task. The mean percentage change between the model's prediction and the counterfactual objective was calculated for the California Housing dataset.

*Task and model scope*: Each task and model.

*Procedure*: The counterfactuals generated were used to alter the original samples and passed to each machine learning model to make predictions. The predictions were then evaluated against each sample's counterfactual target to see whether the counterfactual met its objective. Mean accuracy

was calculated for the Adult Income counterfactuals, and the mean percentage change was calculated for the California Housing counterfactuals.

### A.1.6   6. CERTAINTY

*Objective*: Evaluate the LLM's confidence in its predictions.

*LLM explanation type*: Predicted probabilities.

*Benchmark*: The probabilities predicted by the classifier models.

*Quantitative metric*: The mean RMSE between the LLM-estimated and model-predicted probabilities.

*Task and model scope*: Each classifier model for the Adult Income dataset.

*Procedure*: The RMSE was calculated for each sample between the LLM-estimated and model-predicted probabilities. The mean RMSE was calculated as the LLM's certainty metric.

### A.1.7   7. DEGREE OF IMPORTANCE

*Objective*: Evaluate the LLM's ability to quantify the importance of each feature value on the model's decision.

*LLM explanation type*: Feature importance values.

*Benchmark*: LIME.

*Quantitative metric*: The mean cosine similarity across the LLM feature importance and LIME feature values.

*Task and model scope*: Each task and model.

*Procedure*: The LLM-generated importance values for each sample were compared with their corresponding LIME values, calculating their cosine similarity. Missing values are set as a standard constant of -1 to differentiate them from other values in the dataset. The mean cosine similarity across each ML model was calculated as each LLM's degree of importance score.

### A.1.8   8. COMPREHENSIBILITY

*Objective*: Assess how understandable the LLM-generated explanations are.

*LLM explanation type*: Natural language explanations.

*Benchmark*: No benchmark was available.

*Quantitative Metric*: The mean scores from standard readability tests such as Flesch-Kincaid and Dale-Chall are used as proxies for comprehensibility similar to the method employed by Ali & Kostakos (2023). A full list of readability scores is detailed in Appendix A.4.

*Task and model scope*: Each task and model.

*Procedure*: The readability scores were calculated for each set of LLM-generated natural language explanations. The mean scores were then computed as the LLM's readability metric.

### A.1.9   9. CONSISTENCY

*Objective*: Evaluate how consistent LLM explanations are across different model types.

*LLM explanation type*: Most influential features.

*Benchmark*: No benchmark was available.

*Quantitative metric*: The mean cosine similarity between the feature vectors for each model.

*Task and model scope*: Each task and model.

*Procedure*: Cosine similarity was used to compare each sample's most influential features across all models, with the mean cosine similarity serving as the LLM's consistency score.

### A.1.10   10. STABILITY

*Objective*: Assess how similar the LLM-generated explanations are for identical samples.

*LLM explanation type*: Most influential features.

*Benchmark*: No benchmark was available.

*Quantitative metric*: Average cosine similarity of features from the same input sample.

*Task and model scope*: Each task and model.

*Procedure*: Twenty instances were randomly sampled, repeated five times, shuffled, and reindexed. Each LLM was queried to identify the most influential features for these samples and cosine similarity was calculated between explanations for identical samples. The mean was calculated as the LLM's stability score.

### A.1.11   11. ROBUSTNESS

*Objective*: Evaluate how error-prone the LLM-generated explanations are.

*LLM explanation type*: All available.

*Benchmark*: No benchmark was available.

*Quantitative metric*: The error frequency in the LLM-generated explanations.

*Task and model scope*: Each task and model.

*Procedure*: The error rate was calculated from three perspectives: the number of incorrect samples returned, incorrect columns and rows returned, and invalid outputs, such as all features returned as most influential. The mean error rates were calculated as the LLM's robustness score.

## A.2   RESULTS

| Model | Accuracy | F1 Score | ROC AUC |
|---|---|---|---|
| Random Forest | 0.83 | 0.63 | 0.83 |
| Gradient Boosted Trees | 0.82 | 0.61 | 0.82 |
| KNN | 0.82 | 0.61 | 0.82 |
| gpt-4-turbo | 0.78 | 0.59 | 0.78 |
| Logistic Regression | 0.77 | 0.38 | 0.77 |
| Decision Tree | 0.77 | 0.59 | 0.77 |
| gemma2-9b-it | 0.74 | 0.55 | 0.74 |
| llama3-8b-8192 | 0.69 | 0.41 | 0.69 |
| gpt-4o-mini | 0.67 | 0.54 | 0.67 |
| gemma-7b-it | 0.55 | 0.42 | 0.55 |
| llama3-70b-8192 | 0.55 | 0.48 | 0.55 |

Table 3: Accuracy results for the Adult Income classification task.

| Model | RMSE | MAE | MAPE |
|---|---|---|---|
| Gradient Boosted Trees | 43,159 | 29,374 | 0.16 |
| Random Forest | 45,296 | 30,642 | 0.16 |
| KNN | 61,993 | 42,833 | 0.23 |
| Decision Tree | 62,296 | 42,833 | 0.21 |
| Linear Regression | 70,454 | 51,533 | 0.31 |
| gpt-4-turbo | 199,891 | 162,160 | 1.05 |
| llama3-70b-8192 | 261,863 | 217,909 | 1.51 |
| gemma-7b-it | 273,450 | 242,286 | 1.80 |
| gemma2-9b-it | 295,339 | 243,953 | 1.70 |
| llama3-8b-8192 | 317,304 | 274,996 | 2.12 |
| gpt-4o-mini | 320,902 | 284,693 | 1.87 |

Table 4: Accuracy results for the California Housing regression task.

| LLM Model | Adult Income | California Housing |
|---|---|---|
| llama3-8b-8192 | 0.29 | 0.38 |
| llama3-70b-8192 | 0.29 | 0.38 |
| gpt-4-turbo | 0.24 | 0.43 |
| gemma-7b-it | 0.24 | 0.34 |
| gemma2-9b-it | 0.13 3 | 0.43 |
| gpt-4o-mini | -0.03 | 0.39 |

Table 5: Selectivity results for each task.

| LLM Model | Adult Income | California Housing |
|---|---|---|
| llama3-70b-8192 | 0.28 | 0.23 |
| gemma2-9b-it | 0.08 | -0.60 |
| llama3-8b-8192 | 0.07 | 0.03 |
| gpt-4o-mini | -0.07 | -0.36 |
| gpt-4-turbo | -0.10 | 0.57 |
| gemma-7b-it | -0.21 | -0.63 |

Table 6: Fidelity results for each task.

| LLM Model | Adult Income | California Housing |
|---|---|---|
| gemma-7b-it | -0.03 | -0.08 |
| gpt-4-turbo | -0.13 | -0.04 |
| llama3-70b-8192 | -0.15 | -0.03 |
| llama3-8b-8192 | -0.18 | -0.05 |
| gpt-4o-mini | -0.19 | -0.03 |
| gemma2-9b-it | -0.19 | -0.02 |

Table 7: Completeness results for each task

| LLM Model | Target Accuracy |
|---|---|
| gpt-4-turbo | 0.27 |
| gpt-4o-mini | 0.24 |
| gemma-7b-it | 0.24 |
| llama3-70b-8192 | 0.23 |
| llama3-8b-8192 | 0.23 |
| gemma2-9b-it | 0.23 |

Table 8: Constrastness results for the Adult Income task.

| LLM Model | Mean Percentage Change |
|---|---|
| gemma-7b-it | 0.37 |
| gemma2-9b-it | 0.48 |
| llama3-70b-8192 | 0.51 |
| gpt-4o-mini | 0.64 |
| gpt-4-turbo | 0.73 |
| llama3-8b-8192 | 0.76 |

Table 9: Constrastness results for the California Housing task.

| LLM Model | Mean RMSE |
|---|---|
| gpt-4-turbo | 0.36 |
| gemma2-9b-it | 0.39 |
| gpt-4o-mini | 0.48 |
| llama3-8b-8192 | 0.48 |
| gemma-7b-it | 0.58 |
| llama3-70b-8192 | 0.60 |

Table 10: Certainty results for the Adult Income task.

| LLM Model | Adult Income | California Housing |
|---|---|---|
| gemma-7b-it | 0.11 | -0.20 |
| gpt-4-turbo | 0.03 | -0.13 |
| gemma2-9b-it | 0.02 | -0.05 |
| gpt-4o-mini | -0.01 | -0.08 |
| llama3-8b-8192 | -0.01 | -0.10 |
| llama3-70b-8192 | -0.02 | -0.07 |

Table 11: Degree of importance results for each task

| LLM Model | Fl.-K. | Flesch | Dale-C. | ARI | Lin. W. | Spache |
|---|---|---|---|---|---|---|
| gpt-4-turbo | 11.7 | 45.2 | 9.7 | 11.8 | 13.5 | 7.2 |
| gpt-4o-mini | 10.1 | 53.5 | 8.8 | 9.7 | 11.8 | 6.5 |
| llama3-70b-8192 | 8.8 | 61.6 | 8.1 | 8.4 | 10.8 | 5.9 |
| llama3-8b-8192 | 9.5 | 61.5 | 7.6 | 8.8 | 12.3 | 5.9 |
| gemma2-9b-it | 7.6 | 70.1 | 8.2 | 6.9 | 10.4 | 5.8 |
| gemma-7b-it | 12.1 | 37.8 | 9.4 | 11.5 | 12.5 | 7.0 |

Table 12: Comprehensibility scores for the Adult Income task.

| LLM Model | Fl.-K. | Flesch | Dale-C. | ARI | Lin. W. | Spache |
|---|---|---|---|---|---|---|
| gpt-4-turbo | 12.7 | 38.2 | 10.2 | 13.1 | 14.4 | 7.5 |
| gpt-4o-mini | 10.7 | 48.3 | 10.1 | 10.9 | 12.4 | 6.9 |
| llama3-70b-8192 | 10.5 | 57.3 | 9.5 | 10.6 | 14.3 | 7.2 |
| llama3-8b-8192 | 8.2 | 63.0 | 9.5 | 7.8 | 10.1 | 6.3 |
| gemma2-9b-it | 8.2 | 63.5 | 9.9 | 7.7 | 10.7 | 6.8 |
| gemma-7b-it | 12.4 | 35.2 | 10.5 | 12.2 | 12.0 | 7.7 |

Table 13: Comprehensibility scores for the California Housing task.

| LLM Model | Adult Income | California Housing |
|---|---|---|
| llama3-70b-8192 | 0.84 | 0.91 |
| gpt-4-turbo | 0.81 | 0.95 |
| llama3-8b-8192 | 0.81 | 0.97 |
| gemma-7b-it | 0.80 | 0.85 |
| gpt-4o-mini | 0.77 | 0.94 |
| gemma2-9b-it | 0.69 | 0.93 |

Table 14: Consistency results for each task.

| LLM Model | Adult Income | California Housing |
|---|---|---|
| gpt-4o-mini | 0.99 | 0.92 |
| gpt-4-turbo | 0.64 | 0.81 |
| llama3-70b-8192 | 0.62 | 0.89 |
| gemma2-9b-it | 0.57 | 0.87 |
| gemma-7b-it | 0.57 | 0.87 |
| llama3-8b-8192 | 0.47 | 0.89 |

Table 15: Stability scores for each task.

| LLM Model | Row Error Rate | Column Error Rate |
|---|---|---|
| gpt-4-turbo | 5.00 | 1.09 |
| gpt-4o-mini | 2.86 | 0.00 |
| llama3-70b-8192 | 1.67 | 0.36 |
| llama3-8b-8192 | 4.05 | 0.00 |
| gemma2-9b-it | 1.36 | 0.14 |
| gemma-7b-it | 10.37 | 3.51 |

Table 16: Robustness: Percentage of samples with incorrect rows and columns for the Adult Income dataset.

| LLM Model | Row Error Rate | Column Error Rate |
|---|---|---|
| gpt-4-turbo | 5.00 | 0.0 |
| gpt-4o-mini | 2.50 | 0.0 |
| llama3-70b-8192 | 1.67 | 0.0 |
| llama3-8b-8192 | 3.36 | 0.0 |
| gemma2-9b-it | 1.45 | 0.0 |
| gemma-7b-it | 9.54 | 4.0 |

Table 17: Robustness: Percentage of samples with incorrect rows and columns for California Housing.

| LLM Model | Adult Income) | California Housing |
|---|---|---|
| gpt-4-turbo | 0.00 | 0.00 |
| gpt-4o-mini | 35.65 | 0.54 |
| llama3-70b-8192 | 0.00 | 0.00 |
| llama3-8b-8192 | 0.00 | 5.96 |
| gemma2-9b-it | 3.25 | 0.00 |
| gemma-7b-it | 0.08 | 0.81 |

Table 18: Robustness: Percentage of invalid inputs for each task.

## A.3   PROMPTS

Below is the full list of LLM prompts for each combination of the four input context components (data, role, input, prompt), task, and explanation type.

Data contexts:

- Adult Income: "The features describe aspects of a person and the target labels describe their income status as either low income or high income."

- California Housing: "The features describe the properties of houses in areas within California, USA and the target labels describe the median house prices in those areas."

Role contexts:

- Predictions:
  - Adult Income: "You will receive a dataframe of data describing people where one row is about one person and your role is to predict whether they are low income (0) or high income (1). Provide your answers in the form of a Python list of the same length as the input dataframe with values of 0s or 1s. Do not preface your response with any text. Use your own reasoning and do not use implement code."
  - California Housing: "You will receive a dataframe of data describing the typical price of houses within California, USA. Go through each row in the dataframe and predict the median house price for a particular area. Your response should be in the form of a Python list of the same length as the input dataframe with predicted values in US dollars e.g., 300000.0 for $300k or 500000.0 for $500k. Do not preface your response with any text."

- Predicted probabilities (Adult Income only): "You will receive a dataframe of data describing people where one row is about one person and your role is to predict the probability that they are low income (0) or high income (1). Provide your answers in the form of a Python list of the same length as the input dataframe with values of between 0 and 1. Do not preface your response with any text. Use your own reasoning and do not use implement code."

- Most influential features: "You will receive a dataframe of features and predicted target labels and your role is to identify the most significant influential features that influence the label for each row. Only identify the most influential features and do not include those that are not. There should be at least one feature identified for each row. Do not include every feature in a row. Use your own reasoning and do not use any code such as Python to implement a solution."

- Feature importance values: "You will receive a dataframe of features and predicted target labels and your role is to quantify the importance of each feature in each row with a real number. Each feature show have a number and they should not all be zero. Use your own reasoning and do not use any code such as Python to implement a solution."

- Linear model coefficients:
  - Adult Income: "You will receive features and a predicted target label and your role is to generate the coefficients of a logistic regression model for each feature. The coefficient value should be a real number and you must generate a value for each feature. Do not include a coefficient value for the target label column 'income'. Use your own reasoning and do not use any code such as Python to implement a solution."
  - California Housing: "You will receive features and a predicted target label and your role is to generate the coefficients of a logistic regression model for each feature. The coefficient value should be a real number and you must generate a value for each feature. Do not include a coefficient value for the target label column 'income'. Use your own reasoning and do not use any code such as Python to implement a solution."

- Marginal contributions: "You will receive a dataframe of features and predicted target labels and your role is to quantify the marginal contribution of each feature in each row with a value between 0 and 1. The value for each feature should be greater than 0. Use your own reasoning and do not use any code such as Python to implement a solution."

- Counterfactuals: "You will receive a dataframe of features and predicted target labels and your role is to generate a counterfactual by making a minimal change to the features to change the label. Use your own reasoning and do not use any code such as Python to implement a solution."

- Natural language explanation: "You will receive features and a predicted target label, and your role is to generate a detailed plain English explanation that enables a non-technical layperson to understand how the input features influenced the predicted target label. Use your own reasoning and do not use any code such as Python to implement a solution."

Input contexts:

- Predictions: f"Make {len(X_samples)} predictions for input feature data and ensure there is exactly {len(X_samples)} elements in the Python list. Your response should be only the Python list of predicted values. Do not preface your response with any text."

- Explanations: f"Features and target label dataframe: {sample_data}. Feature column names: {sample_data.columns.tolist()}. Target label column: {target_column}."

Prompt contexts:

- Most influential features: "Return the identified most influential features in the form of a Python dictionary of dictionaries. The outer dictionary's keys should be the index for each sample. There should an inner dictionary for each row with keys as most influential feature names and values as the most influential feature values. Do not include features that are not the most influential. Do not include the target label column. Only return the Python dictionary of dictionaries. Do not explain your solution in any way and do not include any text such as the word Python in your response before or after the Python dictionary."

- Feature importance values: "Return the identified feature importance values in the form of a Python dictionary of dictionaries. The outer dictionary's keys should be the index for each sample. There should an inner dictionary for each row with keys as feature names and values as the feature importance values. Include a non-zero value for each feature. Do not include the target label column. Only return the Python dictionary of dictionaries. Do not explain your solution in any way and do not include any text such as the word Python in your response before or after the Python dictionary."

- Linear model coefficients: "Return the coefficient values in the form of a Python dictionary. The dictionary's keys should be the name of each feature and the dictionary's values should be the feature's coefficient value. Do not include the target label column 'income'. Do not include the target label column. Only return the Python dictionary. Do not explain your solution in any way and do not include any text such as the word Python in your response before or after the Python dictionary."

- Marginal contributions: "Return the identified marginal contribution values in the form of a Python dictionary of dictionaries. The outer dictionary's keys should be the index for each sample. There should be an inner dictionary for each row with keys as feature names and values as the marginal contribution values. Include a non-zero value for each feature. Do not include the target label column. Only return the Python dictionary of dictionaries. Do not explain your solution in any way and do not include any text such as the word Python in your response before or after the Python dictionary."

- Counterfactuals:

  - Adult Income: "Generate a counterfactual for every row in the dataframe that would change the person's income status changing from either low income to high income or high income to low income. The counterfactual must use feature values that exist in the full dataset that are describe in this dictionary of possible feature values {income_possible_features}. Provide the features that should be changed in the form of a Python dictionary of dictionaries. The outer dictionary's keys should be the index for each sample. There should an inner dictionary for each row with keys as names of the features that should be changed and values as the value the feature should be changed to. Only include the features that should be changed. Do not include the target label column. Only return the Python dictionary of dictionaries. Do not explain

your solution in any way and do not include any text such as the word Python in your response before or after the Python dictionary."

– California Housing: "Generate a counterfactual for every row in the dataframe that would increase the median house price for that area by between 20% and 40%. The counterfactual must use feature values that exist in the full dataset that are described in this dictionary of possible feature values {housing_possible_features}. Provide the features that should be changed in the form of a Python dictionary of dictionaries. The outer dictionary's keys should be the index for each sample. There should an inner dictionary for each row with keys as names of the features that should be changed and values as the value the feature should be changed to. Only include the features that should be changed. Do not include the target label column. Only return the Python dictionary of dictionaries. Do not explain your solution in any way and do not include any text such as the word Python in your response before or after the Python dictionary."

• Natural Language Explanations: "Return the explanations in the form of a Python dictionary. The dictionary's keys should be the index for each row and its values should be the row's explanation. Do not include the target label column. Only return the Python dictionary. Do not explain your solution in any way and do not include any text such as the word Python in your response before or after the Python dictionary."

## A.4 READABILITY MEASURES

**Flesch-Kincaid**: The Flesch-Kincaid readability score is a simplified version of the Flesch Reading Ease score. It is designed to estimate a document's US grade level by calculating the average number of syllables per word and the average sentence length in the assessed document. The formula is: US Grade Level $= 0.4 \times average\_sentence\_length + 12 \times average\_syllables\_per\_word - 15$ (Štajner et al., 2012)

**Flesch**: The Flesch Reading Ease score provides a numeric score ranging from 1 to 100 rather than grade-level and also uses the average number of syllables per word and the average sentence length in the assessed document. A low score indicates the document is hard to read. The formula is: Score $= 206.835 - (1.015 \times average\_sentence\_length) - (84.6 \times average\_syllables\_per\_word)$ (Štajner et al., 2012).

**Dale-Chall**: The Dale-Chall readability score provides a numeric score of the readability of a document by assessing how many complicated versus non-difficult words the document contains. The non-difficult comes from a list of 3000 words assumed to be understandable to young American children, with any word not on the list considered difficult. The formula is: 0.1579 (difficult_words / words) * 100 + 0.0496 (words / sentences) (Ley & Florio, 1996).

**Automated Reading Index (ARI)**: The Automated Reading Index (ARI) is designed to assess the reading difficulty of a document and uses the average word and sentence length. Its formula is: 4.71 (characters / words) + 0.5 (words / sentences) - 21.43 (Kincaid & Delionbach, 1973).

**Linsear-Write**: Linsear Write was designed to estimate the US grade level of a document uses an assessment of the number of easy and hard words in the document (Eltorai et al., 2015). It uses the following algorithm:

1. Add one point for each word with two syllables or less

2. Add one point for each word with three syllables or more

3. Add three points.

4. Divide the total points by the number of sentences in the document.

5. If the provision score is greater than 20, divide it by two. Otherwise, divide it by two, then subtract one.

**Spache**: The Space readability method estimates the US grade level of a document by analysing the average sentence length and percentage of unique unfamiliar words. The formula is: Grade Level = 0.121 * average_sentence_length + 0.083 * percentage_of__unfamilar_words + 0.659 (Spache, 1953).

