# OpenReview forum: "Large Language Models for Explainability in Machine Learning"
_ICLR.cc/2025/Conference — Submitted to ICLR 2025_

### Official Review · Reviewer_8fMf · 2024-11-02

**Soundness:** 2
**Presentation:** 3
**Contribution:** 2
**Rating:** 5
**Confidence:** 3

**Summary:**

This work has experimentally evaluated explanations generated by 6 LLMs across 11 dimensions of explainability.

**Strengths:**

- Collating the different dimensions along which one should evaluate explanations in one place is useful.
- Initiating discussions/evaluations on the topic of using LLMs themselves to generate explanations of other ML models and attempting to figure out where LLMs might fit within an explanation system is useful.

**Weaknesses:**

Both the experiments as well as the broader discussion on where LLMs might fit within an explanation system are currently limited. Authors have considered a classification and a regression problem and have attempted to evaluate it deeply. On the discussions front, it would help to assess how LLMs might fit broadly within an explanation system (e.g. conversation, customisation, etc.) beyond simply focusing on computing the explanations themselves.

**Questions:**

1. Trust / black-box issues: Explanations are supposed to transparently explain the behavior of models. However wouldn't using LLMs or other opaque models raises the issue of when one could trust the explanations themselves.

2. Would it not be valuable to discuss more broadly about the different places where LLMs might fit within an explanation system beyond computing the explanations. E.g. talkToModel (https://www.nature.com/articles/s42256-023-00692-8.pdf), customisation, etc.

---

### Official Review · Reviewer_pnT1 · 2024-11-03

**Soundness:** 2
**Presentation:** 3
**Contribution:** 2
**Rating:** 5
**Confidence:** 4

**Summary:**

The authors investigate the quality of LLM generated explanations for two classic tabular data machine learning problems (a) a classification problem (Adult data set) and (b) a regression problem (California Housing data set). Explanation quality has been evaluated along 11 dimensions, among them metrics which typically are used to evaluate explanations such as fidelity, and comprehensibility. Furthermore, predictive accuracy of the LLMs has been evaluated. For the investigated dimensions, specific well-known XAI methods have been used for comparison with LLM generated explanations, for instance, completeness have been evaluated by comparison with SHAP values, selectivity by comparison with DiCE counterfactuals. Evaluation scores have been obtained by cosine similarity metrics for most of the dimensions; comprehensibility of LLM generated explanations has been evaluated by  Flesch-Kincaid readability score. Results show that LLM generated explanations often lack accuracy, are prone to errors, and fail to offer meaningful insights into the inner workings of models.

**Strengths:**

* The authors address a novel and interesting research question -- can LLMs generate explanations for machine learned models which fulfill quality standards which have been established in XAI research -- most important fidelity, robustness and comprehensibility.
* The paper is well written and it is easy to follow.
* The authors motivate their evaluation strategy in a plausible way and explicitly address short comings of the presented evaluation.

**Weaknesses:**

* The evaluation is restricted to learned models for tabular data. In my opinion, for such data, the challenge for creating natural language explanations which are easy understandable to humans is not very high. One possibility would be, to first use a standard XAI methods such as SHAP or DiCE and than use an LLM to generate a short verbal description. However, there is no conclusive evidence from psychology, as far as I know, that verbal explanations are more helpful than the standard output of such XAI methods (for non ML experts such as domain experts or end-users).
* Comparing LLM output to specific XAI methods, in my opinion, would be only a meaningful evaluation measure, if in addition, the respective XAI methods had been evaluated first themselves with respect to the usual quality dimensions (most important faithfulness). How might the results change if the baseline XAI methods were first evaluated for faithfulness? Could you suggest ways to incorporate this evaluation into the study design?

**Questions:**

* Enhancing XAI with LLMs is an interesting research topic, but using tabular data and classic ML approaches such as decision trees or linear regression, seems for me not the best choice to address this question. Such models are interpretable (and not black box) and therefore, explanations of why a model returns a specific class or value can be often given directly from the model application. Enhancing interpretable models with additional XAI methods have been proposed and you use established methods such as SHAP values and DiCE counterfactuals in your evaluation study. Nevertheless, in my opinion, it is an open question whether LLM generated verbal explanations are more helpful to humans than local explanation from the interpretable models themselves or using SHAP etc. Can you give arguments why you think that explanations for interpretable models learned from tabular data are an interesting choice to investigate LLM generated explanations? How might the results differ if applied to more complex, less interpretable models or non-tabular data? What are the potential drawbacks of using interpretable models for this evaluation?

* You present a careful evaluation of explanation quality of LLM generated explanations along 11 dimensions. When you obtain quality metrics for LLMs by comparing their similarity to explanations generated by XAI methods like SHAP values or DiCE counterfactuals, in my opinion, it would have been crucial to first evaluate the quality of these methods themselves. What are your arguments to use use these methods without evaluating their quality as baseline? How would you propose evaluating the quality of the baseline XAI methods before using them as benchmarks? What impact might this have on the interpretation of your results?

---

### Official Review · Reviewer_9gjs · 2024-11-03

**Soundness:** 2
**Presentation:** 1
**Contribution:** 2
**Rating:** 3
**Confidence:** 3

**Summary:**

This work aims to evaluate whether LLMs are capable of providing explanations consistent with or comparable to established explainability criteria. The authors define (?) several orthogonal metrics of evaluation to compare LLM explanations and traditional XAI explanations. XAI methods used are LIME, SHAP, and DiCE Counterfactuals, as well as simpler prediction outputs and probabilities. The authors find that currently, LLMs show limited utility as explainers compared to traditional methods (though better benchmarks are needed), but still have promise as post-hoc explainers.

**Strengths:**

Clear introduction with specifically identified knowledge gap and important research goal: evaluating LLM-generated explanations against established explainability criteria – definitely of interest to the field. Attempts to define distinct evaluation criteria and make important connections between LLM capabilities and established XAI methods. Appendix contains details of experiments. Some salient and significant results, especially about LLMs underperforming on generating counterfactuals, capturing underlying reasoning of model decisions, and producing less accurate predictions than simpler classification models.

**Weaknesses:**

Most weaknesses are regarding clarity and organization, plus care in performing experiments.

Literature survey: the long list of 17 metrics is presented in a confusing way. Is this list compiled by the authors? If so, its positioning in the literature survey is confusing and it should be made clear that this is a list curated by the authors. If it was previously presented by someone compiling the ideas from previous work, whoever compiled it should be clearly cited. Clarity is lacking.

Furthermore, some of the definitions are unclear. E.g. Novelty sounds like a binary classification of whether a sample is novel, while Accuracy is “the ability of explanations to predict novel samples” – does this mean to *say they are novel* (binary classify?), does it mean *correctly predict outputs* for novel samples? It may make sense to define the list you actually use elsewhere (i.e. the one that includes the definition for Robustness, which you say you add – where can I currently see your official definition for Robustness? It should be given in the same format as the others, maybe in the Research Design section.) Overall, I found the organization of the paper weak, which led to this type of confusion.

The definitions are also inconsistent because some are about the produced “explanations,” while others are about the “explanatory method”.

In the explanatory method selection, you don’t give the 1:1 mapping of how you paired LLM-generated explanations with a similar benchmark method for comparison (it’s given later in the results section, but buried in the text - a table could help here).

Usually, a few grammatical errors in the main text of a paper are understandable in an international research community where many are non-native English speakers. However, I see serious problems (consistency, grammar, formatting) with your actual definitions and LLM prompts, making me worry that the work was sloppily performed.
There is inconsistency in your prompt wording for no apparent reason: “Use your own reasoning and do not use implement code.” (ungrammatical) vs. “Use your own reasoning and do not use any code such as Python to implement a solution.” Also “there should an inner dictionary” (ungrammatical) vs. “There should be an inner dictionary.” Hopefully an LLM would not be too affected by these differences, but it is sloppy. I hope these errors are only in your Appendix typesetting and not in your actual study.

Typo line 075 - “ability of human’s how an AI system…” – I assume you mean something like “ability of humans to understand how an AI system…”

The “future work” paragraph of the discussion is not particularly related to the authors’ own findings, though the paper abstract claims they are connected.

This is a relatively small scope project with mainly a negative result. However, it is also hard to conclude that LLMs are incapable of aligning with XAI (hard to fully believe the negative result in a study that was not carefully done). A very carefully designed and described study that found the same results would be an improvement.

**Questions:**

Is the long list of 17 metrics presented in the literature survey compiled by the authors? Are these mostly your final definitions?

Why are you using the Adult dataset? Yes, the task is simple, but both Adult and CA Housing are very old datasets. Adult has a number of problems detailed in the well known NeurIPS Ding et al. 2021 paper, “Retiring Adult: New Datasets for Fair Machine Learning”: https://proceedings.neurips.cc/paper/2021/file/32e54441e6382a7fbacbbbaf3c450059-Paper.pdf. If you are already aware of the identified problems with the Adult dataset (especially with the $50K binary income threshold), I would like to know your justification for still using it; perhaps this information should be in the paper. (Also, it may be interesting for you to investigate whether the explanations align at all with the known limitations of the Adult dataset–but this is out of scope for my review.)

Are there more visual ways of explaining your method or displaying your results? It could help make your paper easier to follow.

---

### Official Review · Reviewer_4fmb · 2024-11-04

**Soundness:** 2
**Presentation:** 2
**Contribution:** 1
**Rating:** 1
**Confidence:** 4

**Summary:**

This paper presents a systematic evaluation of Large Language Models (LLMs) for explainable AI (XAI) by assessing their ability to generate explanations against established properties of explainability. The authors evaluate six prominent LLMs (including GPT-4, LLaMA 3, and Gemma models) across two standard machine learning tasks - Adult Income classification and California Housing regression. They develop a comprehensive experimental framework examining eleven key XAI properties: accuracy, selectivity, fidelity, completeness, contrastness, certainty, degree of importance, consistency, stability, robustness, and comprehensibility. The evaluation uses conventional ML models and XAI methods like LIME, SHAP, and DiCE as benchmarks. The results reveal that while LLMs can generate human-comprehensible explanations, they struggle with crucial aspects like identifying influential features, maintaining consistency, and providing accurate model-behavior representations. The authors conclude that LLMs may be better suited as "translators" of conventional XAI methods rather than primary explainers.

**Strengths:**

- The evaluation spans eleven different XAI properties, providing a holistic assessment of LLM capabilities rather than focusing on narrow aspects.
- The work provides actionable insights about how LLMs could be effectively integrated into XAI systems as translators rather than primary explainers.

**Weaknesses:**

1. The paper's core methodology suffers from severe statistical inadequacy. Using only 1% of test sets (261 samples for Adult Income and 207 for California Housing) violates fundamental statistical principles for reliable model evaluation. For example, with the Adult Income dataset having 14 features and binary classification, 261 samples cannot adequately cover the feature space to evaluate explanation quality. Standard statistical power analysis suggests that for reliable feature importance evaluation with 14 variables, a minimum sample size of ~2000 would be needed (assuming medium effect size and 0.8 power level). The lack of statistical significance testing further compounds this issue - there's no way to determine if the reported differences between LLM and conventional methods are meaningful or just random variation. This is particularly problematic for metrics like stability, where only 20 instances are used, making it impossible to draw statistically valid conclusions about LLM explanation consistency.

2. The paper's evaluation framework has fundamental conceptual flaws in how it measures explanation quality. For instance, using cosine similarity between LLM outputs and LIME/SHAP values as a metric for "completeness" and "degree of importance" is theoretically unsound. LIME and SHAP themselves are approximations with known limitations - LIME's local approximations can be unstable and SHAP values can be misleading when features are correlated. Using these as ground truth creates a circular evaluation where LLM explanations are considered "good" if they match potentially flawed approximations. The paper acknowledges these limitations in Section 2.1 but still uses them as benchmarks without justification. A more valid approach would require:

- Formal theoretical framework for what constitutes a "complete" explanation
- Multiple complementary metrics beyond simple vector similarity
- Ground truth from domain experts or causal analysis
- Evaluation of explanation correctness, not just similarity to other methods


3. The paper fails to provide critical implementation details needed for reproducibility. Specifically:
- ML Model Configurations: No hyperparameters or training details provided
- LLM API Settings: Missing temperature, top-p, and other crucial parameters
- Property Measurement Implementation: No code or detailed algorithms for metric calculations
- Data Preprocessing: Unclear handling of categorical variables and scaling

The prompts shown in Appendix A.3 lack crucial details about how they were developed, tested, or optimized. Without these details, it's impossible to validate the results or build upon this work.

5. The paper lacks fundamental theoretical justification for using LLMs as explainers. No formal framework is presented for:
- Why language models should be capable of understanding and explaining ML model decisions
- How LLM architectures relate to explanation generation
- What properties a valid explanation should satisfy
- How to verify explanation correctness

6. The paper seems to ignore almost all of previous works in this space with the most prominent and closest work being "Kroeger, Nicholas, et al. "Are Large Language Models Post Hoc Explainers?." arXiv preprint arXiv:2310.05797 (2023).".

**Questions:**

N/A

---

### Author Response · Authors · 2024-11-24
**Overall Response to Reviewers**

We thank all reviewers for dedicating their time and effort to providing valuable feedback on our work. We deeply appreciate the insights and constructive suggestions, which will guide us in refining our study.

We wish to address and clarify potential misconceptions, which make it challenging to extract uniformly actionable insights in some areas. The goal of paper is not to conclude that LLMs are fundamentally unsuitable for XAI as we believe this paper supports their potential. The goal is to call for a more rigorous consideration of how LLM-generated explanations of ML models should be evaluated. We believe this is important because of the myriad of ethical, legal and societal reasons that we have mentioned. To this end, we have proposed a framework for evaluating LLM-generated explanations across a diverse range of dimensions of explanation quality. We do not believe this is exhaustive but this framework and the results we have presented serve as an interesting starting point and development with respect to the growing body of literature in this space.

For example, both Reviewers  4mfb and 8fMf question the theoretical justification for why LLMs might generate meaningful explanations. For different reasons, they are both questioning the suitability of LLMs for XAI tasks. In contrast, Reviewer 9gjs expresses scepticism toward the negative results of our study, supporting that LLMs should align with XAI. While a theoretical justification would indeed be interesting, it would be insufficient to conclude that LLMs are suitable for XAI as empirical results would still be required, which in turn requires an evaluation framework. This is the challenge we address and our contribution is to propose an initial framework in which LLM-generated explanations of ML models can be rigorously evaluated.

Our study tackles this question systematically by evaluating LLMs across eleven distinct, imperfect but relevant, metrics that characterize "good" explanations, providing an empirical profile of their strengths and limitations relative to established explainability criteria. To our knowledge, this is among the most comprehensive evaluations of LLM explainability to date. While we do not provide a conclusive theoretical justification for why LLMs might inherently generate high-quality explanations, our study offers rigorous numerical evidence about their current capabilities.

Once again, we thank all reviewers for their thoughtful critiques. We are committed to addressing the identified weaknesses and enhancing the clarity, rigour, and scope of our study. We look forward to further discussions and guidance on these important topics.

---

### Meta-Review · Program_Chairs · 2024-12-24

**Metareview:**

PC is entering the meta-review on behalf of SAC and AC:

 The reviewers felt that the submission was not a strong contribution, and the authors did not respond during the review process.

**Additional Comments On Reviewer Discussion:**

TBD

---

### Decision · Program_Chairs · 2025-01-22

Reject